# Betulonic Acid Inhibits Type-2 Porcine Reproductive and Respiratory Syndrome Virus Replication by Downregulating Cellular ATP Production

**DOI:** 10.3390/ijms251910366

**Published:** 2024-09-26

**Authors:** Feixiang Long, Lizhan Su, Mingxin Zhang, Shuhua Wang, Qian Sun, Jinyi Liu, Weisan Chen, Haihong Wang, Jianxin Chen

**Affiliations:** 1Guangdong Provincial Key Laboratory of Veterinary Pharmaceutics Development and Safety Evaluation, Guangzhou 510642, China; longfx@scau.edu.cn (F.L.); lz-su@stu.scau.edu.cn (L.S.); mxzhang@jnu.edu.cn (M.Z.); shwang@stu.scau.edu.cn (S.W.); qsun@stu.scau.edu.cn (Q.S.); liujinyi1202@163.com (J.L.); 2College of Veterinary Medicine, South China Agricultural University, Guangzhou 510642, China; 3Department of Biochemistry and Genetics, La Trobe Institute for Molecular Science, La Trobe University, Melbourne, VIC 3086, Australia; weisan.chen@latrobe.edu.au; 4College of Life Sciences, South China Agricultural University, Guangzhou 510642, China

**Keywords:** porcine reproductive and respiratory syndrome virus (PRRSV), betulonic acid (BA), anti-PRRSV activity, adenosine triphosphate (ATP)

## Abstract

Porcine reproductive and respiratory syndrome (PRRS), caused by PRRS virus (PRRSV) infection, has been a serious threat to the pork industry worldwide and continues to bring significant economic loss. Current vaccination strategies offer limited protection against PRRSV transmission, highlighting the urgent need for novel antiviral approaches. In the present study, we reported for the first time that betulonic acid (BA), a widely available pentacyclic triterpenoids throughout the plant kingdom, exhibited potent inhibition on PRRSV infections in both Marc-145 cells and primary porcine alveolar macrophages (PAMs), with IC_50_ values ranging from 3.3 µM to 3.7 µM against three different type-2 PRRSV strains. Mechanistically, we showed that PRRSV replication relies on energy supply from cellular ATP production, and BA inhibits PRRSV infection by reducing cellular ATP production. Our findings indicate that controlling host ATP production could be a potential strategy to combat PRRSV infections, and that BA might be a promising therapeutic agent against PRRSV epidemics.

## 1. Introduction

Porcine reproductive and respiratory syndrome (PRRS), caused by porcine reproductive and respiratory syndrome virus (PRRSV) infection, manifests as reproductive failure in sows and as severe respiratory disease and stunted growth in piglets and growing pigs, posing a significant threat to the global swine industry [1]. PRRSV is a small, enveloped RNA virus with a linear, single-stranded, and positive-sense genome, belonging to the family Arteriviridae in the order Nidovirales [2]. Currently, vaccination is the primary method for controlling PRRSV infections. However, the continuous emergence of new strains and restoration or even enhancement of virulence in attenuated vaccines due to host adaptation and recombination present challenges in PRRSV control [3]. Therefore, there is an urgent need to develop novel antiviral strategies against PRRS outbreaks. Previous studies have identified several natural compounds with antiviral activity against PRRSV infection, including andrographolide [4], artesunate [5] and toosendanin [6]. However, none of these reported compounds has been demonstrated clinically, and there is no clinically effective anti-PRRSV drug available.

Natural compounds, including various plant-derived ones like alkaloids, terpenes, phenols, and flavonoids, have a well-documented efficacy against viral infections [7]. Betulonic acid (BA, Figure 1A), a pentacyclic lupane-type triterpenoid, is widely distributed throughout the plant kingdom. BA has drawn increasing attention due to its diverse biological and pharmacological activities, including antibacterial, antimalarial, antidiabetic, antitumoral, and general anti-inflammatory effects [8]. Recent studies have shown that the anti-cancer mechanism of BA is associated with its ability to inhibit the adenosine triphosphate (ATP) production of cancer cells [9]. BA was reported to have antiviral activities, including on influenza A virus (IAV), herpes simplex type 1 virus (HSV-1), and human immunodeficiency virus 1 (HIV-1) [10,11]. However, little is known about antiviral mechanism of BA, and whether BA inhibits PRRSV infection is unknown.

Adenosine triphosphate (ATP), the major energy currency in all living cells, has long been known to be essential for various intracellular processes [12]. Viruses utilize host cell organelles and synthesis machinery to facilitate their replication. Most viruses fulfill their replication using host ATP supply [13,14]. For example, HIV-1 infection induces cellular ATP release through pannexin-1-mediated interaction between the HIV-1 envelope protein and specific target cell receptors, which in return promotes HIV-1 replication [15]. White spot syndrome virus (WSSV) hijacks shrimp autophagy for lipid utilization and ATP production to provide energy for its replication [16]. Our previous study showed that the antimalaria drug artesunate inhibits PRRSV replication by activating 5′-adenosine monophosphate (AMP)-activated protein kinase (AMPK) by increasing AMP and ADP levels while decreasing ATP levels in infected Marc-145 cells and primary porcine alveolar macrophages [5]. However, little is known about the dependence of PRRSV replication on host cellular ATP production in the infected cells.

In the present study, we show that BA potently inhibits PRRSV infection at micromolar concentrations in a dose-dependent manner in Marc-145 cells and PAMs. Mechanistically, we demonstrate that BA’s inhibition of PRRSV infection is associated with its suppression of cellular ATP production. Since BA has been extensively reported as an anti-cancer compound with excellent pharmacological properties and minimal toxicity, our work provides valuable insight into its potential applications in controlling PRRSV infection.

## 2. Results

### 2.1. BA Inhibits PRRSV Replication in Marc-145 Cells

The cytotoxicity of BA was initially assessed using an MTT assay. The results showed that BA at concentrations of less than 20 µM did not affect Marc-145 cell viability, and its CC_50_ value was determined to be 47.5 µM (Figure 1B). Subsequently, the inhibitory effects of BA against three different PRRSV strains (GD-HD, CH-1a, and NADC30-like) were evaluated using IFA. As depicted in Figure 1C–F, BA significantly inhibited PRRSV replications in a dose-dependent manner. The IC_50_ values of BA against the three tested PRRSV strains were determined to range from 3.3 to 3.7 µM by analyzing infected cell percentage in IFA images. Ribavirin, a well-known inhibitor of viral RNA polymerase, was employed as a positive antiviral drug control in this study. Our results indicated that 160 μM of ribavirin exhibited significant inhibition against the three tested PRRSV strain infections in the same assays. The sustained inhibition of PRRSV replication by BA was evaluated by analyzing the kinetics of viral titer, viral RNA, and N protein expression from 24 to 72 h post-infection (hpi). As depicted in Figure 2A, treatment with BA resulted in a significant dose-dependent reduction in viral titer at all indicated time points. Specifically, treatment with 10 μM of BA led to a 6.1 log reduction in progeny virus production at 72 hpi, compared to that in the DMSO-treated control. The inhibitory effects of BA on PRRSV replication were further corroborated by reduced viral NSP9 RNA levels (Figure 2B) and viral N protein expressions (Figure 2E). In addition, BA treatment exhibited consistent inhibition on PRRSV replication with different challenging virus doses, including 10, 100, and 1000 TCID_50_ (Figure 2C–F). These results demonstrated BA’s sustaining and stable inhibition of PRRSV propagation in Marc-145 cells.

### 2.2. BA Inhibits PRRSV Replication in PAMs

As PAMs represent the primary target cells of PRRSV in pigs, we therefore investigated whether BA also has inhibitory effect on PRRSV replication in PAMs. BA was well tolerated by the PAMs, with an IC_50_ of 5.4 μM for PRRSV replication in PAMs (Figure 3A,B). Treating PRRSV-infected PAMs with 20 μM of BA led to 95% reduction in NP-expressing cells (Figure 3C), consistent with the proportionally decrease in viral N protein mRNA expression (Figure 3D) and virus titers (Figure 3E). These results show that BA significantly inhibits PRRSV replication in PAMs.

### 2.3. BA Inhibits PRRSV Replication by Interfering with the Late Stage of the Virus Replication Cycle Rather than Its Entry

To determine which stage of the PRRSV replication cycle was affected by BA, we conducted a time-of-addition experiment. Marc-145 cells were treated with 10 µM of BA before (pre-treatment), during (co-treatment), or after (post-treatment) PRRSV infection (Figure 4A). The results showed that pre-treatment with BA for 2 h did not change PRRSV N protein expression or virus titer, suggesting that BA did not affect the susceptibility of Marc-145 cells to PRRSV infection. When cells were treated with BA during PRRSV infection (co-treatment), PRRSV N protein expression and virus titer did not decrease (Figure 4B–D), indicating that BA did not directly inactivate the virions or interfere with PRRSV entry. Notably, BA treatment for 24 h after PRRSV infection significantly reduced PRRSV N protein expression and virus titer (Figure 4B–D), suggesting that BA exerts inhibitory effects on PRRSV replication in Marc-145 cells post virus entry.

To further identify the specific stage in which BA exerts its anti-PRRSV effect during a PRRSV replication cycle, a time course experiment was performed as shown in the diagram in Figure 5A after infection. The results showed that BA exerted significant inhibition on PRRSV replication when added at 9–12 hpi (Figure 5B,C), while BA addition at 0–9 hpi did not inhibit PRRSV replication, suggesting that BA interferes with the late stage of the PRRSV replication cycle, such as the assembly and release of virions, rather than virus entry or an early replication stage. Meanwhile, ribavirin exerted its antiviral effect when added at 0–3 hpi, which is consistent with its reported role in interfering with viral RNA synthesis [3].

### 2.4. BA Inhibits PRRSV Replication by Reducing ATP Production

Most viral replication processes require ATP as an energy source. However, as viruses are unable to produce their own energy supply, they rewire cellular metabolic pathways to obtain energy and other necessary resources from the infected host cell. BA has been reported to reduce ATP production by suppressing ADP phosphorylation [17]. Thus, we hypothesized that BA might exert its anti-PRRSV effects by decreasing cellular ATP production. To verify this hypothesis, we investigated the dynamic effects of PRRSV infection and BA treatment on the cellular ATP levels in Marc-145 cells at 3, 6, 9, 12, 24, and 48 hpi. The results showed that PRRSV infection significantly increased the cellular ATP level at 12 hpi but reduced the ATP levels at 24 and 48 hpi, compared to the mock control (without BA treatment or PRRSV infection). Meanwhile, for Marc-145 cells without PRRSV infection, BA treatment consistently reduced the ATP levels at 9, 12, 24 and 48 h, compared to the mock control. For PRRSV-infected cells, BA treatment reduced the ATP level at 12 hpi but positively regulated the ATP levels at 24 and 48 hpi, compared to the virus control (Figure 6A). These results collectively indicated that PRRSV infection induces the upregulation of cellular ATP production at the earlier stage (12 hpi), followed by the decrease in ATP production at the later stages (24 and 48 hpi). BA’s antiviral activity might be associated with its inhibition of ATP production. To confirm this finding, we examined the effect of exogenous ATP addition on PRRSV replication. Our results showed that the addition of 100 μM ATP increased the production of progeny viruses (Figure 6D). Moreover, exogenous ATP addition significantly reversed BA’s inhibition of PRRSV replication, reflected in the increased percentage of PRRSV-infected cells (Figure 6B,C) and the virus titer in the supernatant (Figure 6D). In contrast, exogenous ATP addition did not reverse ribavirin’s inhibition of PRRSV replication (Figure 6B–D).

To further elucidate the connection between host ATP production and PRRSV infection, we assessed the anti-PRRSV effects of two distinct ATP inhibitors: rotenone and oligomycin A. As depicted in Figure 6E–G, rotenone and oligomycin A significantly inhibited PRRSV replication in Marc-145 cells, reflected by reduced infected cells (Figure 6E,F) and viral titer (Figure 6G), in a dose-dependent manner. Not surprisingly, the addition of exogenous ATP reversed the inhibitory effects of both rotenone and oligomycin A on PRRSV replication. These results confirmed that PRRSV replication relies on ATP supply from the host cells, and inhibition of ATP production leads to attenuated PRRSV replication.

## 3. Discussion

PRRSV infection causes enormous financial losses to the swine industry worldwide. PRRSV-vaccinated or convalescent animals are protected against reinfection in largely a homologous, rather than heterologous, PRRSV strain-specific manner. Due to highly mutated PRRSV genes and the multitudinous genotypes of co-circulating PRRSV, the efficiency of commercially available vaccines is limited. Consequently, research aimed at developing new antiviral strategies against PRRSV infection is urgently needed. The present study is the first to show that betulonic acid (BA) potently inhibits PRRSV infection in Marc-145 cells and PAMs, with IC_50_ values ranging from 3.3 to 5.4 μM (Figure 1, Figure 2 and Figure 3). Furthermore, our study shows that BA inhibits PRRSV replication by negatively regulating cellular ATP production at the earlier stage of the virus infection (Figure 6A–D). The robust inhibition of two different ATP inhibitors (rotenone and oligomycin A) on PRRSV replication (Figure 6E–G) confirmed the necessity of ATP supply for PRRSV infection, and controlling ATP levels might be a potential strategy to combat PRRSV infection.

Viruses heavily rely on their host’s energy production mechanisms, mainly ATP synthesis, for their replication [18]. More and more studies have shed light on how viruses actively manipulate and exploit cellular glycolytic and fermentation pathways to fuel their replication, highlighting the crucial role of energy in viral processes [19]. Additionally, viruses are known to strategically interfere with host metabolism to establish persistent infections. For example, infections of Hepatitis B virus (HBV) and Hepatitis C virus (HCV) have been shown to disrupt hepatic metabolic responses, impacting glucose, lipid, nucleic acid, bile acid, and vitamin metabolism [20,21]. HCV infection, specifically, leads to increased expression of glycolytic enzymes during its replication process, and HCV affects ATP-sensitive potassium channels through the glycosyltransferase-mediated N-glycosylation of glucose transporter 2 on the surface of pancreatic β-cells, leading to impaired insulin secretion [22]. In addition, metabolic analyses during herpes simplex virus type 1 (HSV-1) infection have revealed enhanced ATP production, accompanied by increased mitochondrial Ca^2+^ content and proton leakage, causing the extracellular release of ATP [23,24]. Our previous study showed that the antimalaria drug artesunate inhibits PRRSV replication by activating 5′-adenosine monophosphate (AMP)-activated protein kinase (AMPK) by increasing the AMP/ADP:ATP ratio in Marc-145 cells and PAMs [5]. However, little is known about the relationship between the PRRSV infection and host cellular ATP production in virus-infected cells. Pujhari et al. reported that ATP production in PRRSV-infected Marc-145 cells was significantly reduced in the later stages of infection (at 24 and 48 hpi). They further demonstrated that the PRRSV E protein interacts with mitochondrial proteins and induces apoptosis [25]. Nevertheless, the reduction in ATP production at the later stages of PRRSV infection is likely caused by cell apoptosis induced by the virus infection, and the correlation between cellular ATP production with the PRRSV replication in the earlier stage of the virus infection remains to be revealed. In the present study, we observed that PRRSV infection induced a significant increase in cellular ATP level at 12 hpi (Figure 6A), at which point the viruses are in the logarithmic phase of replication and need more energy supply from the host cells [19]. Meanwhile, BA treatment reduced ATP production in PRRSV-infected and mock-infected Marc-145 cells at 12 hpi (Figure 6A). Mahmoudabadi et al. reported that influenza viruses consume different amounts of energy in their replication cycle and that the translation of viral proteins and viral exit are the most energetically expensive processes, meaning that more ATP supply is required at these two points (8–12 hpi) in the virus replication cycle [26]. PRRSV requires a similar time cost to the influenza virus to complete one replication cycle [6], and the translation of viral proteins and viral exit of PRRSV consume more ATP, which may explain why PRRSV infection induced a significant increase in cellular ATP level at 12 hpi in our study. These results collectively suggest that PRRSV replication is dependent on the host ATP-based energy supply, and the anti-PRRSV activity of BA is associated with its suppression of cellular ATP production. The results of the time-of-addition assay showed that BA exhibited anti-PRRSV activity when added during 9–12 hpi; meanwhile, when BA was added before 9 hpi, it did not inhibit PRRSV replication (Figure 4 and Figure 5), which is consistent with the time window of BA’s suppression of ATP production induced by PRRSV infection. Of note, in our study, PRRSV infection led to decreased cellular ATP levels at later stages (24 and 48 hpi) in the virus replication cycle (Figure 6A), in line with Pujhari’s observation [25], which can be attributed to host cell apoptosis during the later stages of PRRSV infection and apoptosis-caused ATP loss. BA exhibited consistent inhibition of ATP production over 9 to 48 h of incubation in Marc-145 cells without PRRSV infection (Figure 6A). However, for PRRSV-infected cells, BA treatment led to reduced ATP level at 12 hpi and a less pronounced decrease in ATP levels at 24 hpi and 48 hpi, compared to the virus group (Figure 6A). It can be explained that BA inhibits PRRSV replication then indirectly decreases PRRSV-induced apoptosis in Marc-145 cells. These results collectively suggest that BA’s inhibition of PRRSV replication is closely associated with its suppression of host ATP production.

Mitochondria are unique double-membraned organelles present in all eukaryotic organisms. Acting as a powerhouse to generate energy, the mitochondria plays essential roles in physiology to sustain life [27]. The electron transport chain (ETC), a key component of the mitochondria, is the main site of ATP production [18]. Within the mitochondrial ETC, electrons liberated from reducing substrates are delivered to O_2_ via a chain of respiratory H^+^ pumps [28]. These pumps (complexes I–IV) establish a H^+^ gradient across the inner mitochondrial membrane, and the electrochemical energy of this gradient is then used to drive ATP synthesis via complex V (ATP synthase). Signal transduction, metabolism, immune response, cell cycle, and apoptosis are all associated with mitochondrial function [29]. BA was reported to disrupt mitochondrial oxidative phosphorylation and biosynthesis processes by targeting mitochondrial NADH dehydrogenase (complex I), finally inhibiting ATP production [9,30]. It can be speculated that BA’s inhibition of ATP production in our study is likely due to its suppression of mitochondrial complex I. However, the observation that BA only exhibited its inhibition of ATP production after incubation with the cells for 9 h and 12 h (Figure 6A) remains to be fully explained. Previous studies have demonstrated that two ETC inhibitors, rotenone (inhibitor of complex I) and oligomycin A (ATPase inhibitor), inhibit HCMV replication by decreasing host ATP production [31]. In our study, rotenone and oligomycin A showed potent inhibition of PRRSV replication in a dose-dependent manner, and exogenous ATP addition robustly reversed the inhibition of PRRSV replication caused by BA, rotenone, and oligomycin A (Figure 6) but did not reverse the antiviral effect of ribavirin. These findings validate the importance of host ATP production during PRRSV replication and indicate that controlling host ATP level might be a promising strategy to combat PRRSV infection. However, whether PRRSV infection affects the enzyme activity or protein expression of ETC complexes and how BA suppresses ATP production by affecting ETC complexes are still unknown and need to be further investigated.

An earlier study reported that intraperitoneal injection of BA (50–500 mg/kg) in mice showed no toxicity [32]. Smith et al. reported that 3-o-(3′, 3′-dimethylsuccinyl) betulonic acid (bevirimat), a BA derivative, exhibited no human toxicity at an oral dose of 250 mg/day (for consecutive 20 days) and successfully passed a phase II clinical trial against HIV infection [33]. These studies confirm the safety of BA in vivo application.

Here, BA caused the inhibition of PRRSV replication in vitro, with effective concentrations ranging from 2.5 to 10 μM (Figure 1, Figure 2 and Figure 3). This raises the question of whether such concentrations of BA could inhibit PRRSV in pigs. Li et al. reported that serum BA levels reached 21.9 μM in rats at 4 h post an intraperitoneal injection of 12 mg/kg [34]. Considering the difference in pharmacokinetics between rats and pigs, it can be estimated that intraperitoneal injection of BA at 2.1 mg/kg in a 40 kg piglet would result in a serum BA concentration of 21.9 μM at 4 h post injection [35]. In our study, BA caused the potent inhibition of PRRSV replication with an IC_50_ of 5.4 µM in PAMs, a significantly lower BA serum concentration than 21.9 μM, meaning that an effective antiviral BA serum concentration is easily achievable clinically. Of note, studies to assess BA’s toxicity and anti-PRRSV effect in pigs are imperative.

## 4. Materials and Methods

### 4.1. Cell Lines and Virus Strains

Marc-145 cells, a PRRSV-permissive cell line derived from African green monkey kidney cell line MA-104, were obtained from the American Type Culture Collection (ATCC) and grown in Dulbecco’s minimum essential medium (DMEM, Gibco, Logan, UT, USA) supplemented with 10% fetal bovine serum (FBS, Biological Industries, Kibbutz Beit Haemek, Israel), 100 IU/mL of penicillin, and 100 µg/mL streptomycin at 37 °C with 5% CO_2_. Porcine alveolar macrophages (PAMs) were obtained from the lungs of 4- to 6-week-old PRRSV-negative Large White piglets (Xinli Pig Farm, Wuzhou, China) by lung lavage according to a previously described method [36]. Briefly, the lungs were washed three times with pre-cooled phosphate buffered saline (PBS) solution containing penicillin (300 IU/mL) and streptomycin (300 µg/mL). Cells were centrifuged at 800× *g* for 10 min, resuspended in RPMI 1640 supplemented with 10% FBS, 100 IU/mL of penicillin, and 100 µg/mL streptomycin at 1 × 10^6^ cells/mL in a 6-well plate, and then incubated at 37 °C for 2 h. The suspending cells were removed, and the adherent cells were PAMs [37].

### 4.2. Antibodies, Chemicals, and Reagents

Betulonic acid (BA) was purchased from Chendu Pufei De Biotech Co., Ltd. (Chendu, China), with a purity of ≥99.3%. Ribavirin, a broad-spectrum antiviral agent, was used as a positive control and purchased from Star Lake Bioscience Co., Ltd. (Zhaoqing, China). Adenosine triphosphate (ATP), rotenone, and oligomycin-A were purchased from MCE China (Shanghai, China). All the compounds were dissolved in dimethyl sulfoxide (DMSO, Sigma-Aldrich, Burlington, MA, USA). BA was diluted with essential medium before use. The final concentration of DMSO in the culture medium was less than 0.4%. Anti-PRRSV N protein mouse monoclonal antibody was purchased from Median Diagnostics (1:1000; Chuncheon, Republic of Korea).

### 4.3. Cytotoxicity Assay

The cytotoxicity of tested compounds on Marc-145 cells or PAMs was evaluated by MTT assay [5]. In brief, Marc-145 cells or PAMs were cultured in 96-well plates for 24 h, followed by incubation with essential medium containing various concentrations of compounds at 37 °C for 48 h. Subsequently, the cells were washed with PBS and incubated with 100 μL 3-(4,5-dimethylthiozol-2-yl)-3,5-dipheryl tetrazolium bromide (MTT; Sigma-Aldrich, MA, USA) solution (0.5 mg/mL in PBS) at 37 °C for 4 h and then added to 150 μL DMSO. The absorbance was measured at 490 nm by a microplate reader (Thermo fisher scientific, Waltham, MA, USA). The value of 50% cytotoxic concentration (CC_50_) was analyzed by GraphPad Prism 8.0 (GraphPad Software, San Diego, CA, USA).

### 4.4. Quantitative Real-Time PCR (qPCR)

Total RNA was extracted from cells or culture supernatants using the total RNA rapid extraction kit (Fastagen, Shanghai, China), following manufacturer’s instructions. RNA was reverse-transcribed into first-strand cDNA using a reverse transcription kit (TaKaRa, Dalian, China) [38]. PCR amplification was performed on 1 µL of reverse-transcribed product with primers designed against PRRSV-NSP9 and GAPDH (glyceraldehyde-3-phosphate dehydrogenase, used as the endogenous control). The following are the sense and anti-sense primer sequences: PRRSV-NSP9 (5′-CCTGCAATTGTCCGCTGGTTTG-3′ and 5′-GACGACAGGCCACCTCTCTTAG-3′); GAPDH (5′-GCAAAGACTGAACCCACTAATT-3′ and 5′-TTGCCTCTGTTGTTACTTGGAG-3′). Real-time PCR was performed using 2×RealStar Green Power Mixture (containing SYBR Green I Dye) (Genstar, Beijing, China) on the CFX96 Real-time PCR system (Bio-Rad, Hercules, CA, USA). Relative mRNA expression was calculated by 2^−∆∆CT^ method using DMSO-treated infected cells or DMSO-treated mock-infected cells as reference samples for determining PRRSV-NSP9 gene expression, respectively [4].

### 4.5. Indirect Immunofluorescence Assay (IFA)

For immunostaining, the PRRSV-infected and uninfected cells were each fixed with 4% paraformaldehyde for 10 min, permeabilized with 0.25% Triton X-100 for 10 min at room temperature (RT), blocked with 1% bovine serum albumin (BSA) for 60 min at RT, and then incubated with a mouse monoclonal antibody against the N protein of PRRSV (clone 4A5, 1:400 dilution, MEDIAN Diagnostics, Chuncheon-si, Republic of Korea) at 4 °C overnight. After three washes with PBS, the cells were incubated for 1 h at RT with a goat anti-mouse secondary antibody conjugated with Alexa Fluor^®^ 568 (red) (Cell Signaling Technology, Beverly, MA, USA) at a 1:1000 dilution. Nuclei were counterstained using 50 µL of 4,6-diamidino-2-phenylindole (DAPI, 300 nM; Sigma-Aldrich, USA) (blue). Immunofluorescence was captured using the Leica DMI 4000B fluorescence microscope (Leica, Wetzlar, Germany). The relative N protein level (%) of each image was calculated based on the fluorescence optical density (OD) using Software Image J version 1.8.0 (USA). Results from compound-treated samples were compared to those from corresponding DMSO-treated control groups (set as 100%). The IC_50_ value (the concentration required to reduce viral N protein by 50%) was determined by plotting the relative N protein level as a function of compound concentration and calculated using GraphPad Prism 8.0 software (GraphPad Software, San Diego, CA, USA).

### 4.6. Western Blot Analysis

PRRSV-infected and uninfected Marc-145 cells with or without BA treatment were lysed in RIPA lysis buffer containing 1 mM phenylmethylsulfonylfluoride (Beyotime, Haimen, China) at 4 °C. The supernatant was harvested after centrifugation (15,000× *g* for 10 min at 4 °C), and the total protein for each sample measured using the BCA protein assay kit (Beyotime, Haimen, China). Ten micrograms of total protein per sample was electrophoresed onto a 12% SDS-PAGE gel and transferred to polyvinylidene-fluoride (PVDF) membranes (Millipore, Burlington, MA, USA). After blocking, the membranes were incubated with a mouse anti-PRRSV N protein monoclonal antibody (clone 4A5, MEDIAN Diagnostics, Chuncheon, Republic of Korea) or antibodies against α-tubulin (1:1000; Beyotime, Haimen, China). Membranes were incubated with HRP-conjugated goat anti-mouse or anti-rabbit IgG (H–L) secondary antibodies (1:3000; Beyotime, Haimen, China). The Odyssey system (LICOR, Hartford, CT, USA) was used to analyze the PVDF membranes.

### 4.7. Time-of-Addition Assay

A time-of-addition assay was performed as previously described [36] with some modifications. Marc-145 cells were grown in 24-well plates to confluence and then infected with the PRRSV GD-HD strain at 100 TCID_50_ for 2 h at 37 °C. BA was added before (pre-treatment), during (co-treatment), or after (post-treatment) PRRSV infection. For pre-treatment, cells were incubated with BA for 3 h at 37 °C, followed by three washes with PBS, and then infected with PRRSV for 2 h. For co-treatment, cells were simultaneously incubated with BA and PRRSV at 37 °C. After 2 h, the mixture was removed, and the cells were washed three times with PBS before fresh medium was added. For post-treatment, cells were first infected with PRRSV for 2 h at 37 °C and then incubated in the fresh medium containing BA for 0–3, 3–6, 6–9, or 9–12 h, followed by continuous incubation in fresh medium. At 24 hpi, samples were submitted to determinations of virus titer by endpoint dilution assay and PRRSV-infected cell counting by IFA.

### 4.8. Determination of ATP Content

Marc-145 cells were infected with PRRSV (100 TCID_50_) for 2 h and then treated with BA for 3, 6, 9, and 12 h, and then cells were lysed at 4 °C for 10 min using an ATP assay kit (Beyotime, Haimen, China). The lysate was centrifuged for 5 min (12,000× *g* at 4 °C), and the supernatant was harvested and measured for the total protein level using bicinchoninic acid protein assay kits (Beyotime, Haimen, China). The ATP content in each sample was detected using ATP assay kits (Beyotime, Haimen, China) [5].

### 4.9. Statistical Analysis

All experiments were performed in at least triplicates. All data are shown as mean ± SD. Statistical significance was determined by Student’s *t*-test when only two groups were compared or by one-way analysis of variance (ANOVA) when more than two groups were compared. * *p* < 0.05, ** *p* < 0.01, and *** *p* < 0.001 were considered as statistically significant differences.

## 5. Conclusions

In conclusion, our study reveals that BA exerts a potent inhibitory effect on PRRSV replication in both Marc-145 cells and PAMs. Mechanically, we showed that PRRSV replication relies on energy supply from cellular ATP production, and BA inhibits PRRSV infection by disrupting cellular ATP production. Our findings offer valuable, novel insights for the development of anti-PRRSV agents. Further in vivo investigations are necessary to confirm BA’s potential as an effective PRRSV inhibitor in swine.

## Figures and Tables

**Figure 1 ijms-25-10366-f001:**
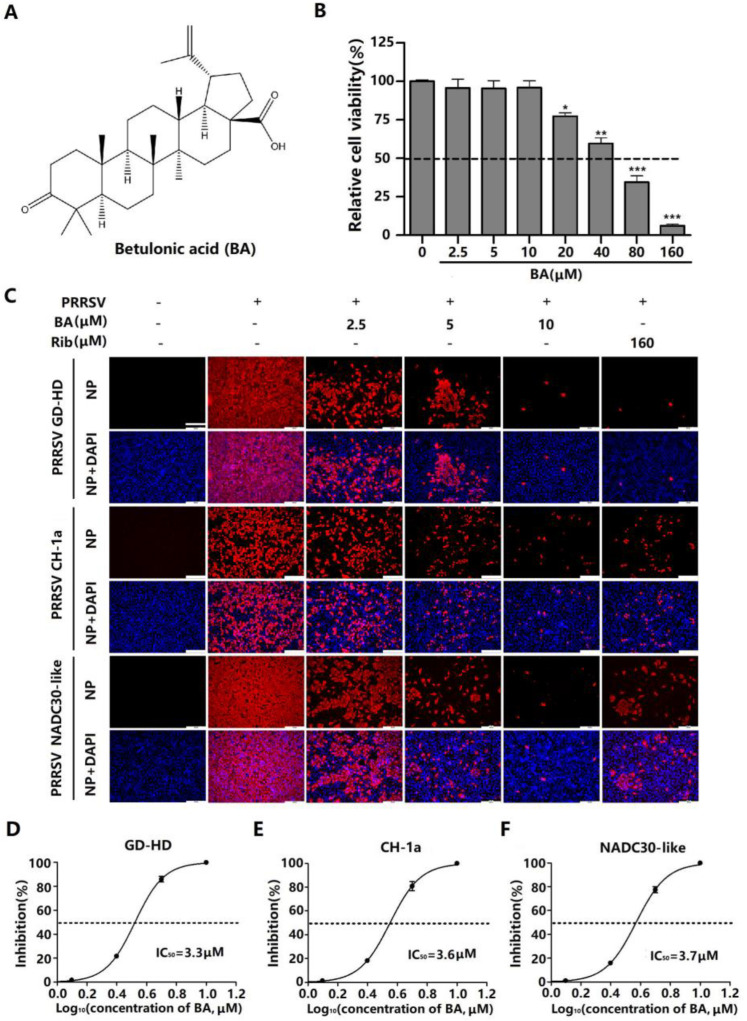
BA inhibits PRRSV replication with minimal cytotoxicity in Marc-145 cells. (**A**) Chemical structure of betulonic acid (BA). (**B**) Cellular toxicity of BA on Marc-145 cells was examined using an MTT assay. (**C**) Antiviral activity of BA against PRRSV strain (GD-HD, CH-1a, and NADC30-like) infections in Marc-145 cells was examined using indirect immunofluorescence assay (IFA). Cells grown in 96-well plates were infected with PRRSV (100 TCID_50_) for 2 h at 37 °C and then cultured in fresh media containing various concentrations of BA. IFA for PRRSV N protein was performed at 48 hpi using Alexa Fluor 568-conjugated goat anti-mouse secondary antibody (red). Nuclei were counterstained using 4,6-diamidino-2-phenylindole (DAPI) (blue). Representative IFA images of three independent experiments are shown in (**C**). (**D**–**F**) show the percentage of inhibition based on the fluorescence optical densities (OD) of the images from three independent experiments. Software Image J was used to digitize image OD. * *p* < 0.05, ** *p* < 0.01, and *** *p* < 0.001 compared to the DMSO-treated control. Scale bar in (**C**): 250 µm.

**Figure 2 ijms-25-10366-f002:**
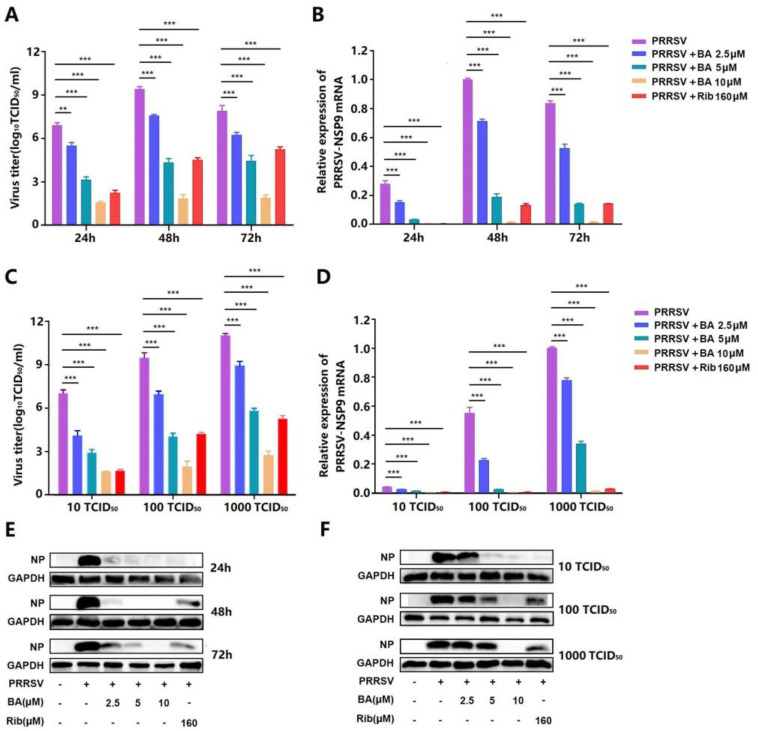
BA shows anti-PRRSV activity in Marc-145 cells. Marc-145 cells grown in 12-well plates were infected with 100 TCID_50_ of PRRSV GD-HD for 2 h and incubated with fresh media containing different concentrations of BA. Samples were collected at 24, 48, and 72 hpi (**A**,**B**), or cells were infected with PRRSV at different doses (10, 100, or 1000 TCID_50_) for 2 h at 37 °C and then treated with different concentrations of BA for 48 h (**C**,**D**). The supernatants were used for determining virus titer using the end-point dilution assay (**A**,**C**), and the cells were used for the analysis of relative PRRSV NSP9 mRNA levels using qRT-PCR (**B**,**D**) and of viral N protein expressions using Western blot (**E**,**F**). Expression of GAPDH was used as a loading control in the Western blots. A DMSO-treated sample (0 μM BA) was used as the control. ** *p* < 0.01, and *** *p* < 0.001 compared to the DMSO-treated control.

**Figure 3 ijms-25-10366-f003:**
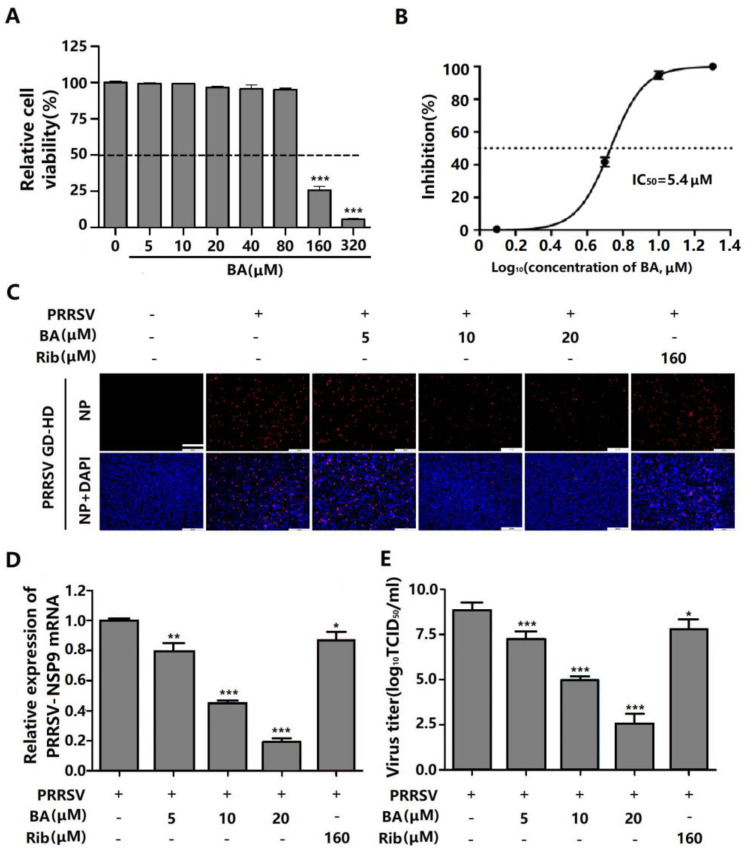
Cytotoxicity and anti-PRRSV activity of BA in PAMs. (**A**) Cellular toxicity of BA in PAMs was examined after 24 h of incubation using the MTT assay. (**B**–**E**) PAMs grown in 24-well plates were infected with GD-HD PRRSV for 2 h at 37 °C and then treated with various concentrations of BA for 24 h. The expression of viral N protein was analyzed by IFA (**C**) and (**B**) shows the percentage of inhibition based on the fluorescence optical densities (OD) of images from three independent experiments. Parallel samples were submitted for analysis of viral NSP9 gene mRNA level using qRT-PCR (**D**) and of viral titer using end-point dilution assay (**E**), respectively. * *p* < 0.05, ** *p* < 0.01, and *** *p* < 0.001 compared to the DMSO-treated control. Scale bar in (**C**): 250 µm.

**Figure 4 ijms-25-10366-f004:**
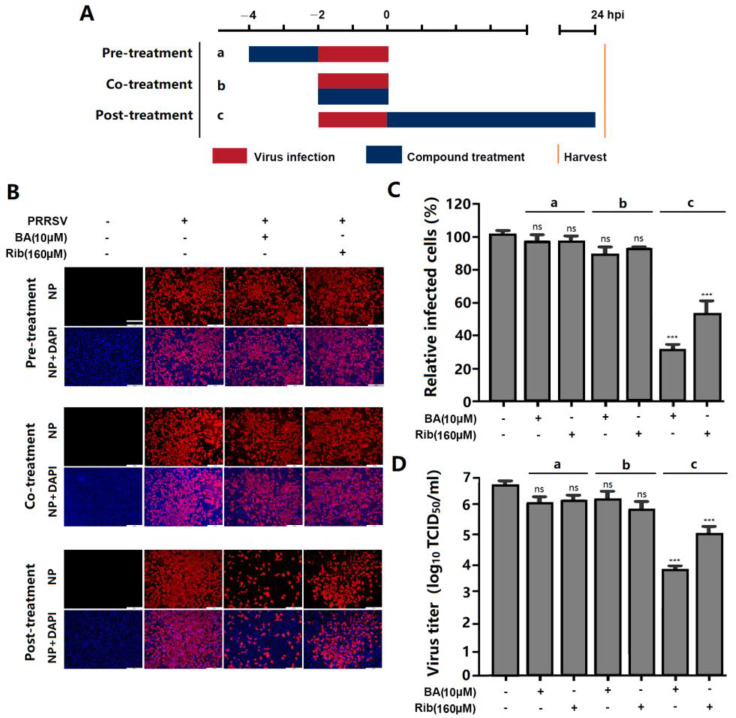
BA inhibits PRRSV replication post-treatment, rather than pre- or co-treatment. Marc-145 cells were infected with GD-HD PRRSV (100 TCID_50_) for 2 h at 37 °C and then washed three times to remove unbound virions. The different BA treatment modes are shown in (**A**). In pre- and post-treatment modes, cells were treated with 10 μM of BA before or after PRRSV infection, respectively. In co-treatment mode, cells were infected with GD-HD PRRSV for 2 h in the presence of 10 μM of BA. At 24 hpi, the cells and supernatants were subjected to viral N protein analysis by IFA (**B**,**C**) and virus titration analysis by end-point dilution assay (**D**), respectively. *** *p* < 0.001, ns means no significant, compared with the corresponding viral control. Scale bar in (**B**): 250 µm.

**Figure 5 ijms-25-10366-f005:**
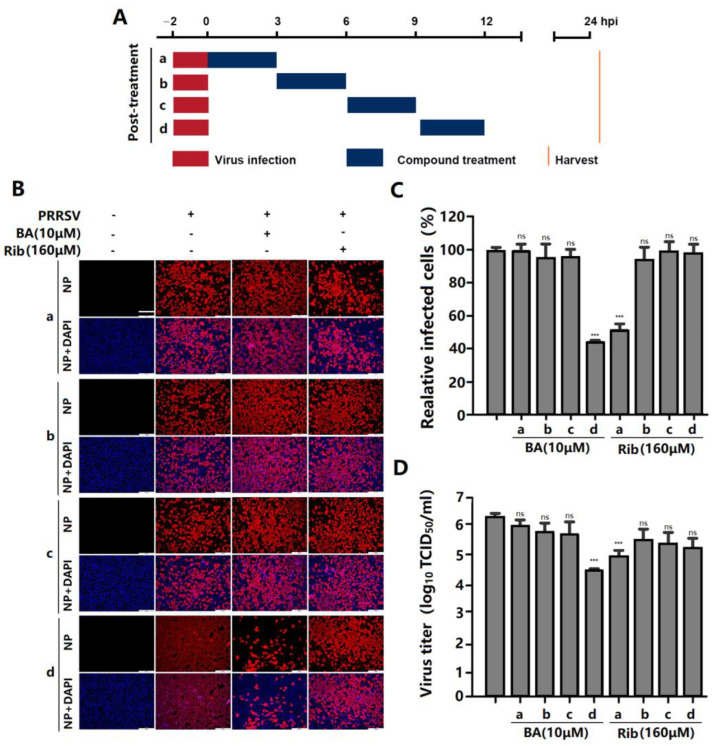
BA inhibits PRRSV infection at the late stage of the viral replication cycle. Marc-145 cells were infected with PRRSV GD-HD (100 TCID_50_) for 2 h at 37 °C and washed three times to remove unbound virions, and then 10 μM of BA was added into the 0–3, 3–6, 6–9, and 9–12 hpi cell groups (**A**). At 24 hpi, the cells and supernatants were subjected to an analysis of viral N protein by IFA (**B**,**C**) and of virus titer by end-point dilution assay (**D**), respectively. *** *p* < 0.001, ns means no significant, compared with the corresponding viral control. Scale bar in (**B**): 250 µm.

**Figure 6 ijms-25-10366-f006:**
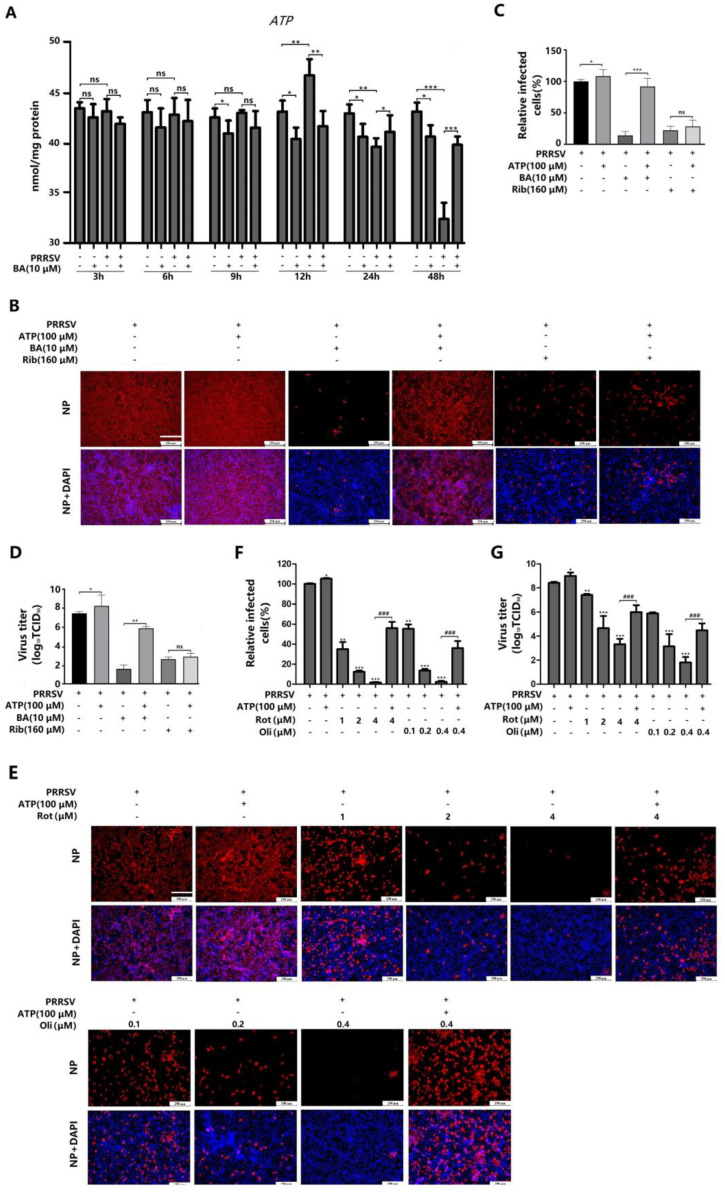
BA inhibits PRRSV replication by suppressing ATP production. Marc-145 cells were infected with PRRSV GD-HD (100 TCID_50_) for 2 h at 37 °C and then washed three times to remove unbound virions, and then the cells were treated with fresh media containing 10 μM of BA for 3, 6, 9, or 12 h, followed by analysis of their ATP levels in the cell cultures (**A**). Marc-145 cells were infected with PRRSV GD-HD (100 TCID_50_) for 2 h at 37 °C and then treated with 10 μM BA, 160 μM Rib, or 100 μM ATP alone or 10 μM BA or 160 μM Rib combined with 100 μM ATP (**B**–**D**). At 24 hpi, relative PRRSV-infected cells were determined by IFA (**B**,**C**), and the supernatants were collected for detecting virus titer (**D**). Marc-145 cells were infected with PRRSV GD-HD (100 TCID50) for 2 h at 37 °C and then treated with different concentrations of rotenone (Rot) and oligomycin A (Oli) for 24 h, with 100 μM ATP alone, or with 100 μM ATP combined with Rot or Oli (**E**–**G**). At 24 hpi, relative PRRSV-infected cells were determined by IFA (**E**,**F**), and the supernatants were collected for detecting virus titer (**G**). * *p* < 0.05, ** *p* < 0.01, *** *p* < 0.001, ^###^ *p* < 0.001, ns means no significant, compared with the corresponding viral control. Scale bar in (**B**,**E**): 250 µm.

## Data Availability

All data related to this study are included in the article. Other data can be obtained from the corresponding author if reasonably required.

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
