# Peer review of "Betulonic Acid Inhibits Type-2 Porcine Reproductive and Respiratory Syndrome Virus Replication by Downregulating Cellular ATP Production"

_ijms, 2024, doi:10.3390/ijms251910366_

Round 1
Reviewer 1 Report
Comments and Suggestions for Authors
Overall, really good. just a few minor revisions. see attached.

Reviewer 2 Report
Comments and Suggestions for Authors
The authors present here a detailed research into the inhibition on PRRSV replication conducted by a triterpenoid compound through the down-regulating of cellular ATP production, with effective concentrations of 2.5 to 10 μM. The toxicity of compound BA was also evaluated to be higher than 20 μM. Control experiments clearly demonstrated the MOA of ATP regulation. Overall, this is a well-writen paper suitable for publication
The authors described here research into triterpenoid compound Betulinic acid (BA) on its activity against porcine reproductive and respiratory syndrome virus, then a plausible mechanism was proposed and also evaluated to prove its reliability. So a new promising strategy against porcine reproductive and respiratory syndrome virus was evaluated together with a mechanism which may also guide future directions of the analog design and optimization. Yes, the topic is relevant to this field, where researcher dive into the discovery of new active compounds against various microbes with novel mechanisms. This research here provides researchers with the opportunity to apply a simple triterpenoid compound to combat the porcine reproductive and respiratory syndrome virus with good activity (IC50 ranging from 3.3 μM to 3.7 μM), which could be used to develop the clinical treatment that can save lives in the future. The present form is suitable for publication, for future improvement, the authors may consider further optimization of the chemical structure to hopefully further enhance the activity. And in vivo tests may also be helpful to prove the effectiveness. The conclusions discussed here are well demonstrated and correspond well with the results obtained. And they do address the question the authors posed in the beginning. Yes, references are appropriate.Tables and figures looks good and did a good job in explaining the work has been done. Comments on the Quality of English Language
Good
